# Sexual Abuse vs. Sexual Freedom? A Legal Approach to the Age of Sexual Consent in Adolescents in Spanish-Speaking Countries

**DOI:** 10.3390/ijerph181910460

**Published:** 2021-10-05

**Authors:** Sandra M. Parra-Barrera, María del Mar Sánchez-Fuentes, Carlos Fuertes-Iglesias, Miguel Ángel Boldova

**Affiliations:** 1Department of Criminal Law, Philosophy of Law and History of Law, University of Zaragoza, 50009 Zaragoza, Spain; abogadasandraparra@gmail.com (S.M.P.-B.); cfuertes@unizar.es (C.F.-I.); mboldova@unizar.es (M.Á.B.); 2Department of Psychology and Sociology, Faculty of Human and Social Sciences, University of Zaragoza, 44003 Teruel, Spain; 3Department of Social Sciences, Faculty of Human and Social Sciences, University de la Costa, Barranquilla 080002, Colombia

**Keywords:** child sexual abuse, age of sexual consent, Romeo and Juliet clause, sexual freedom, sexual indemnity, comparative law

## Abstract

Child and adolescent sexual abuse (CSA) is an international public health problem. Despite the importance of CSA, there is no consensus definition, and the lack of consensus is related to difficulties in conducting prevalence studies as well as research in other areas. To establish a consensual definition, legal aspects such as the age of sexual consent and the difference in age or power between victim and aggressor as well as aspects related to sexual freedom and sexual indemnity must be considered. Therefore, the main goal of this research was to analyze the age of sexual consent in the legal systems of Spanish-speaking countries and to examine whether the Romeo and Juliet clause is established. To achieve the proposed aims, we employed the legal interpretation method, and we analyzed the current Criminal Codes of the 21 Spanish-speaking countries. From the results, it is found that the age of sexual consent varies between countries, establishing valid sexual consent between 13 and 18 years. In addition, only six countries have the Romeo and Juliet clause that protects sexual freedom in adolescents. Finally, we discussed the lack of consensus on the age of sexual consent and the limitations presented by the Romeo and Juliet clause.

## 1. Introduction

Child and adolescent sexual abuse (CSA) is considered a public health problem worldwide, not only affecting the victims but also their families and society [1,2,3,4]. Some of the most common consequences of CSA include physical health problems such as sexually transmitted infections, injuries, and unwanted pregnancies [5,6]; and mental health problems such as anxiety and depression [7,8], post-traumatic stress disorder [9], attention deficit hyperactivity disorder [10,11], personality disorders [11], and severe mental disorders such as schizophrenia [12].

CSA is also considered a public health problem as the numbers are staggering. However, all the available data are mostly based on approximations, highly dependent on the countries studied, the adopted definition of CSA, the methodology used, or the groups evaluated [13]. Nonetheless, it is estimated that between 10% and 12% of children and adolescents are victims of sexual abuse [14,15,16,17]. Among them, approximately 120 million girls under the age of 20 are reported to have suffered CSA [18], the incidence of which is considerably higher in low- and middle-income countries [19]. These trends are best exemplified in Latin America, where an estimated 1.1 million girls have been victims of sexual violence or any other forced sexual act since childhood [20].

Despite the importance of the CSA, there is no agreed definition of this phenomenon. On the one hand, the Centers for Disease Control and Prevention (CDC) have conceptualized CSA as the attempts and/or having carried out sexual acts, contact, and/or sexual exploitation of a child [21]. On the other hand, the World Health Organization defined it as “the involvement of a child in sexual activity that he or she does not fully comprehend, is unable to give informed consent to, or for which the child is not developmentally prepared, or else that violates the laws or social taboos of society. Children can be sexually abused by both adults and other children who are—by virtue of their age or stage of development—in a position or responsibility, trust or power over the victim” [22]. The lack of consensus from establishing a definition of CSA is often related to the complexity of quantitatively assessing the phenomenon, which is a quandary that could potentially derive in the underestimation of its actual numbers. To develop a common definition that can be agreed upon by the legal community, it is necessary to consider health-related issues as well as our conceptualization of childhood and adolescence and their social and legal ramifications [23]. In addition, according to the WHO, it is a challenge not only to establish what constitutes abuse—or even determine what is understood as childhood and adolescence—but also to consider the differences in age or power between the victim and the perpetrator [24]. In this regard, previous investigations have concluded that one of the most frequent types of CSA involves victims under the age of 13 and aggressors under the age of 18 (both of which are considered minors), often with a five-year difference between them [25,26].

From a legal standpoint, Criminal Law finds the minimum age of sexual consent to be an essential aspect of the study and analysis of CSA. It is relevant to point out that the minimum age to recognize the validity of sexual consent is both historically and socially conditioned. Historically, certain legal systems did not consider sexual relations between adults and adolescents a crime, since they were understood as having a reproductive function rather than being necessarily hedonistic [27]. However, legal systems have undergone transformations due to the development of societies, changes in the understanding of sexuality from a much broader point of view and not focused on procreation [27], the recognition of a legal framework to protect children and adolescents [28,29], and the orientation of sexual crimes toward the protection of sexual freedom [30]. Age of sexual consent means “the age below which, in accordance with national law, it is prohibited to engage in sexual activities with a child” [31]; therefore, it is the legislative power of the State who determines the minimum age of sexual consent.

Two essential factors in the consideration of CSA and sexual crimes are sexual freedom and sexual indemnity, both of which are protected legal rights. On the one hand, since sexual and reproductive rights are manifestations of the free development of the personality, they are considered fundamental rights and, therefore, universal [32]. Thus, sexual rights are related to personal freedom and autonomy, privacy, and recognition of human dignity. Children and adolescents are individuals with fundamental rights such as equality, dignity, and free development of the personality, which must be respected. However, the recognition of sexual freedom in the case of minors, and especially regarding the age of initiation of sexual activity with third parties, is usually limited legally and by social and cultural aspects [33]. As such, the exercise of rights in each vital stage may vary, and their protection should always be promoted through the legal institutions provided for the representation or assistance of minors as appropriate, recognizing their developmental capacity and progressively greater autonomy [28,30]. For this reason, it is common for the legal framework to establish a minimum age of sexual consent, limiting the capacity for self-determination. Another relevant aspect is the existence of the well-known ‘Romeo and Juliet’ clause, which is an exemption for cases in which adolescents have consented to have sexual relations with a person close to them by age and/or degree of development and maturity [34]. This clause is essential to avoid considering sexual activities among adolescents who have not yet reached the minimum age of sexual consent as criminal offenses. Otherwise, the sexualized and natural behavior of adolescent individuals would be penalized. However, the doctrine has also indicated that it is a necessary clause but with defects in its statement.

Sexual indemnity can sometimes interfere with sexual freedom, since sexual freedom is directly related to sexual self-determination, and minors lack their own self-determination or are limited by the legislator when establishing a minimum age to grant validity to sexual consent, as well as the right to freely develop the personality in the sexual sphere [35]. Therefore, sexual freedom is related to the assumptions of valid sexual consent, while sexual indemnity refers to the assumptions in which there is no legal capacity to consent and so that, despite giving said consent, it would not be valid. Sexual indemnity refers to not suffering sexual harm or the right not to suffer it [36], that is, sexual indemnity is responsible for the protection of minors in relation to sexual behaviors that may negatively influence their development. It is considered that sexual relations between adults and minors, in which there is a difference in age and therefore also a difference in evolutionary maturity, are practices that can harm the minor, since they would be abusive sexual practices in which they would be at a disadvantage [37].

Given the lack of consensus on a definition on CSA, the possible differences in the consideration of the minimum age to establish sexual consent in minors, and the high prevalence of CSA in Spanish-speaking countries as well as the absence of investigations that have carried out a review of the legal systems of these countries in relation to the minimum age of sexual consent, the main goal of this research is to analyze the legal system regarding the age of sexual consent in minors in Spanish-speaking countries. In addition, a specific aim is proposed: to examine if the legal system of Spanish-speaking countries establishes any ‘Romeo and Juliet’ clause.

## 2. Materials and Methods

### 2.1. Study Design

We use the method of legal interpretation because it is the most appropriate approach to analyze contradictions and/or possible deficiencies in the norms [38].

### 2.2. Materials

The materials used were the Criminal Codes currently enforced in all Spanish-speaking countries where Spanish is the official language, in particular, the articles referring to the age of sexual consent in minors. Table 1 shows the Criminal Codes analyzed.

### 2.3. Procedure

First, two of the researchers located the Criminal Codes of 21 Spanish-speaking countries and selected the articles referring to the age of sexual consent in minors. These legal systems can be consulted online. In order to ensure the reliability of the study, two additional researchers completed the same process independently. This procedure was carried out between the months of May and June 2021.

## 3. Results

After analyzing the articles referring to sexual crimes in the Criminal Codes of the 21 Spanish-speaking countries, we found that the minimum age established in minors regarding the validity of their sexual consent ranged from 13 (in Argentina [39]) to 18 (Dominican Republic, Ecuador and Equatorial Guinea [45,46,47]). The age mode regarding sexual consent in minors is 14 years old, age established in ten of the 21 countries analyzed [40,41,42,43,49,50,52,53,54,55]. This legal age of sexual consent would be enforced by almost half of the countries studied (47.6%), followed by those who set it at 16 years old (19%) [44,56,57,59], at 15 (14.3%) [48,51,58], at 18 (14.3%) [45,46,47], and finally at 13 years old (4.8%) [40]. In addition, Chile and Paraguay present differences in the legal age of sexual consent for heterosexual and homosexual behaviors. Whereas the age of sexual consent for heterosexual behaviors is set at 14 years old in both cases, the legal age of consent for their homosexual counterparts varies a year (17 in Paraguay and 18 in Chile [41,54]). These results are presented in Table 2.

We also examined whether in the Criminal Codes of the Spanish-speaking countries selected there were any articles referring to the exclusion of criminal responsibility in minors that would illustrate the typicality of conduct in certain cases of sexual interaction with minors of the age of consent. An example of this would be the ‘Romeo and Juliet’ clause, which refers to the difference in age and/or degree of development of maturity among minors for the consideration of a criminal offense related to CSA [34]. Only six countries (28.6%)—of the 21 analyzed—presented a ‘Romeo and Juliet’ clause, which is a considerable contrast with the remaining 15 countries (71.4%) where this normative consideration does not exist (see Table 2).

In Bolivia, there is a ‘Romeo and Juliet’ law modality in the case of the following assumption: (a) the relationship must be consensual, without violence or intimidation; (b) the members must be adolescents over 12 years of age; and (c) the maximum age difference between them must be 3 years. In addition, there is also an excuse for acquittal—different from the atypical clause based on consent—when “the defendants marry the victims, provided there is free consent, before the sentence that is enforceable.” In this case, we are dealing with a kind of post-criminal behavior that affects the sphere of punishable behavior [40].

Although in Costa Rica, the ‘Romeo and Juliet’ clause does not expressly exist, it can be inferred from the criminal regulation, where some frameworks permit sexual relations with minors depending on the age of the active subject and always under the assumption of the consent of the minor: (a) if the victim is between 13 and 15 years old and the age difference does not exceed five years with the active subject, such conduct is not sanctioned and (b) if the victim was over 15 years old and less than 18, and the difference of age between the perpetrator and the minor does not exceed seven years, in which case this conduct will not be penalized [43].

In Panama, a type of ‘Romeo and Juliet’ clause is established, “when there is a duly proven permanent relationship between the victim and the aggressor and provided that the age difference does not exceed five years” [53].

In Puerto Rico, exemption from the penalty is also established in the case of sexual relations that take place between persons over 14 years of age, where the maximum age difference between the perpetrator and the victim is four years [56].

In Spain, art. 183 quarter contemplates the ‘Romeo and Juliet’ clause. This article specifies that free consent of the minor under 16 years of age will exclude criminal responsibility for the crimes foreseen, except in the case of article 183.2 of the Penal Code that refers to sexual acts in which there is violence and intimidation, when the perpetrator is a person close to the minor by age and degree of development or physical and psychological maturity [57].

Finally, in Uruguay, a presumption of violence in sexual acts is established when they are carried out with persons under the age of 15. However, an exception for such a presumption is presented if it involves “consensual relationships between persons of 13 years of age or older” and between those where there is no “difference of age greater than eight years” [58].

## 4. Discussion

CSA is a public health problem, with serious consequences for both victims and societies [1,2,3,4] and with a high prevalence worldwide that is more accentuated in low- and middle-income countries [18,19]. The minimum age of sexual consent in minors is not only an essential factor for the consideration of sexual crimes but also a key to carry out research on their prevalence, consequently improving the current definition of CSA. Since the minimum age of sexual consent is determined by each country, this article was aimed at researching different Spanish-speaking legal systems to see how they differed with regard to this topic—as well as with their adoption of a ‘Romeo and Juliet’ clause. After the analysis of the Criminal Codes, we concluded that the minimum age of sexual consent varied among the countries studied, suggesting that the comparison of CSA figures across Spanish-speaking countries should be carried out with caution.

In our analysis, which included the Criminal Codes of 21 Spanish-speaking countries, we verified that in comparative law, it is common to organize age groups according to the different evolutionary stages of childhood and adolescence, establishing the victims of the different types of sexual crimes, as well as the aggravating circumstances. While the minimum age of sexual consent varies between countries, 14 is the most frequent age established across almost half of the sample [40,41,42,43,49,50,52,53,54,55]. However, ranges spanned from 13 [39] up to 18 years old [44,45,46,47,48,51,56,57,58,59]. The ages considered in the different types of crimes are established by the judicial power of each of the states without attending to a universal and logical criterion of treatment of the minor in their sexual protection [60].

The age of sexual consent refers to the age below which, according to national legislation, it is forbidden to carry out sexual activities with a child, even though the child allows it [31]. As mentioned above, the age of sexual consent has been fundamentally increased by not only considering sexual intercourse for its reproductive function [27] but in accordance with the protection of the rights of children and adolescents [28,29] and the protection of their sexual freedom and sexual indemnity [30]. However, there is no consensus on what the specific age should be. On one hand, it is considered that setting the age of consent at 16 years old or more may pose a problem for the free development of the personality and the enjoyment of sexual and reproductive rights [61]. Furthermore, it is important to note that previous research has concluded that the average age for initiating sexual intercourse is between 14 and 16 years [62]. Therefore, countries that establish 16 years or more as the valid age of sexual consent criminalize sexual behaviors that are part of natural development. On the other hand, there are organizations that defend the need to raise the age of sexual consent. For example, in Spain, the original age of sexual consent established was 12 years [63] but was later increased to 13 years old in 1999 [64] and further raised to 16 years old in 2015 [58]. These changes were made in accordance to the recommendations of the UN Committee on the Rights of the Child [65], which considered that the age of 13 was very low for sexual consent and could pose a greater risk for sexual exploitation of children and adolescents. In the case of Latin America, the Ibero-American Convention on the Rights of Young People establishes that individuals over 15 years of age have the right to freely choose a partner and to live together [66]. Therefore, countries that establish an age greater than 15 years for sexual consent have not considered this Convention in their legislation.

Our analysis confirms that reaching a consensus regarding the age of sexual consent is a difficult task that responds to the lack of clear rules in the field of international law. In addition, legal systems regarding crimes of sexual violence must protect both sexual freedom and sexual indemnity. Therefore, the consideration of the ‘Romeo and Juliet’ clause could be fundamental in protecting legal rights, freedom, and sexual indemnity. This clause excludes criminal liability for sexual crimes as long as adolescents have consented to have sexual relations with a person close to them in age [35], and in the case of Spain, also in the degree of development or physical and psychological maturity [58]. Notwithstanding, the above mentioned is not without problems. First, only six of the 21 Spanish-speaking countries include a ‘Romeo and Juliet’ clause in their legal order [40,43,53,56,57,58]. Second, the age established as valid for the consideration of sexual consent in said clause varies, ranging from 13 (in Bolivia, Costa Rica and Uruguay) [40,43,58] to 16 (in Spain) [58]. Third, the age difference for consideration of a crime exemption also varies between countries: including differences no greater than three years (Bolivia) [41], four (Puerto Rico) [56], five (Costa Rica and Panama) [43,53], seven years for adolescents between 15 and under 18 years (Costa Rica) [43], and, at most, eight years (Uruguay) [58]. In the case of Panama, an age of consent is not established in this clause, but for the exemption to occur, there must be a permanent relationship between the individuals [53]. For example, in the case of Spain, the existence of age proximity (rather than a specific difference) and a similar degree of development of their physical and psychological maturity is considered [57].

Although the ‘Romeo and Juliet’ clause protects underage active subjects in some cases of sexual crimes by safeguarding their sexual freedom, it is not exempt from limitations that may be promoted by CSA. On the one hand, there is the age difference of the subjects. For example, in the case of Uruguay [58], a 20-year-old young adult may have sexual relations with a 13-year-old adolescent without any criminal liability or being considered an instance of CSA. However, the differences in the physical and psychological development of individuals between these ages are considerable [67] to the extent that the adult could take advantage of the minor. As a result, instead of protecting sexual freedom, the legislation would ignore/not protect the legal interest of sexual indemnity. Another example would be the case of Spain, where the legislator does not propose a specific age difference but rather the existence of age and developmental proximity. This can put legal security at risk, since it would be the one administering justice who would make a subjective assessment regarding the chronological, physical, and psychological proximity of the individuals involved. In turn, this could result in different judicial pronouncements according to the prejudices, education, and beliefs of the judge [68].

Another important aspect to take into account is the presence of discriminatory legislation based on gender and sexual orientation in relation to the age of sexual consent. In Chile and Paraguay, for example, 17–18 are the established ages for valid sexual consent to participate in homosexual sexual acts [41,54]. This consideration constitutes discrimination for the LGBTI community, criminalizing homosexual sexual relations given the lack of equality based on sexual orientation, which would, in turn, violate their fundamental rights.

### Limitations

The main limitation of this investigation is its scope, as it was restricted to the analysis of the current Criminal Codes of Spanish-speaking countries. For this reason, future research based on comparative law should go beyond Spanish-speaking countries and analyze both the age of sexual consent and the ‘Romeo and Juliet’ clause at the international level instead. In addition, retrospective studies would also be of interest, focusing not only on current Criminal Codes but also on previous ones, verifying how the age of sexual consent has changed in criminal law. In addition, we recommended future studies examining similarities and differences between Spanish-speaking countries and other countries.

## 5. Conclusions

This study constitutes the first time that the legal system of 21 Spanish-speaking countries is analyzed with the aim of documenting the age of sexual consent and the existence of a ‘Romeo and Juliet’ clause, which are both essential in considering the definition and prevalence of CSA.

Sexual freedom and sexual indemnity are legal interests protected in CSA crimes. On the one hand, it is necessary to protect the free development of the personality and, therefore, the practice of sexual relations with/between adolescents. On the other, it is essential to protect the sexual indemnity of children and adolescents, because suffering CSA has serious consequences. For this reason, establishing the age of sexual consent in the legal system is a key issue when dealing with crimes of sexual violence. However, the lack of homogeneity regarding this determination shows the absence of a decisive criterion at the international level.

Countries have an obligation to protect minors from sexual abuse, and they should also safeguard sexual freedom, at least among adolescents who are close in age. Therefore, the lack of a norm that ignores sexual consent among adolescents could be considered unconstitutional because certain minors are being denied their fundamental rights [62]. The ‘Romeo and Juliet’ clause hereby examined protects sexual relations between minors, although, as shown, it is not without problems. Therefore, legislators should work alongside professionals from other sectors to continue analyzing this clause in order to overcome its current limitations.

## Figures and Tables

**Table 1 ijerph-18-10460-t001:** Laws that have modified sexual crimes in the Criminal Codes of Spanish-speaking countries.

Country	Criminal Code
Argentina	Law 25087 [39]
Bolivia	Law 2023 [40]
Chile	Law 19.617 [41]
Colombia	Law 1236 [42]
Costa Rica	Law 9406 [43]
Cuba	Law 87 [44]
Dominican Republic	Law 2497 [45]
Ecuador	Law 180 [46]
Equatorial Guinea	Law [47]
El Salvador	Decree 210 [48]
Guatemala	Law 9-2009 [49]
Honduras	Decree 234-2005 [50]
Mexico	Federal Criminal Code [51]
Nicaragua	Law 846 [52]
Panama	Law 21 [53]
Paraguay	Law 6002 [54]
Peru	Law 31040 [55]
Puerto Rico	Law 34 [56]
Spain	Law 1/2015 [57]
Uruguay	Law 19580 [58]
Venezuela	Law 38668 [59]

**Table 2 ijerph-18-10460-t002:** Age of sexual consent and existence of the Romeo and Juliet laws in the legal-criminal system of Spanish-speaking countries.

Country	Age of Sexual Consent	Romeo and Juliet Laws
Argentina	13 years old	No
Bolivia	14 years old	Yes (art. 308 bis y 317)
Chile	14 years old (heterosexual behaviors); 18 years old (homosexual behaviors.)	No
Colombia	14 years old	No
Costa Rica	14 years old	Yes (art. 159)
Cuba	16 years old	No
Dominican Republic	18 years old	No
Ecuador	18 years old	No
El Salvador	15 years old	No
Equatorial Guinea	18 years old	No
Guatemala	14 years old	No
Honduras	14 years old	No
Mexico	15 years old	No
Nicaragua	14 years old	No
Panama	14 years old	Yes (art. 176 in fine CP)
Paraguay	14 years old (heterosexual behaviors); 17 years old (homosexual behaviors)	No
Peru	14 years old	No
Puerto Rico	16 years old	Yes (art. 130)
Spain	16 years old	Yes (art. 183 quater CP)
Uruguay	15 years old	Yes (272 BIS.1)
Venezuela	16 years old	No

## Data Availability

Not applicable.

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
