# Peer review of "Sexual Abuse vs. Sexual Freedom? A Legal Approach to the Age of Sexual Consent in Adolescents in Spanish-Speaking Countries"

_ijerph, 2021, doi:10.3390/ijerph181910460_

Round 1

Reviewer 1 Report

line. 36 - take out 'even' line 51. Should attempt have an s on it? I think it makes more sense without line 52 - defines line 59 - 'from' not 'in' line 63 - take out 'and' line 81 - last word that not who line 10. Is 'assumptions' in which there is not legal capacity the right word? IS it the situation where there is no capacity? I'm not sure I'm right - but think about what word goes here. This doesn't seem correct line 204 - take out second 'and' line 208. As this crime as a whole' doesn't make sense. I think finish the sentence at CSA. line 209 'country, not ; line 228 - consents to it (add to) line 228-232. I've changed two words here - it might not be what you mean....As mentioned above, the age of sexual consent has been fundamentally increased by not only considering sexual intercourse for its reproductive function [28], but in accordance with the protection of the rights of children and adolescents [29, 30] and the protection of their 231 sexual freedom and sexual indemnity line 233 - take out 'the' - on one hand... line 277-278. remove dashes line 284 - result in rather than derive?

Author Response

The authors greatly appreciate the comments and suggestions made by the reviewer. 

Comment 1. line. 36 - take out 'even'.

Response: Done.

Comment 2. line 51. Should attempt have an s on it? I think it makes more sense without line 52

Response: The authors explained that in addition to the definition of CDC, the definition of line 52 is important, which corresponds to the WHO, since this definition takes into account the concept of consent.

Comment 3. Defines line 59 - 'from' not 'in'.

Response: Done.

Comment 3. line 63 - take out 'and'.

Response: Done.

Comment 4. line 81 - last word that not who

Response: The authors want to state that it is the legislative power of the state that does establish the age limit. That is, the state does establish the minimum age.

Comment 5. line 10. Is 'assumptions' in which there is not legal capacity the right word? IS it the situation where there is no capacity? I'm not sure I'm right - but think about what word goes here. This doesn't seem correct.

Response: The authors confirm that they are "assumptions" and not "situations".

Comment 6. line 204 - take out second 'and'

Response: Done.

Comment 7. line 208. As this crime as a whole' doesn't make sense. I think finish the sentence at CSA.

Response: Done.

Comment 8. line 209 'country, not ;

Response: Done.

Comment 9. line 228 - consents to it (add to)

Response: Done.

Comment 10. line 228-232. I've changed two words here - it might not be what you mean....As mentioned above, the age of sexual consent has been fundamentally increased by not only considering sexual intercourse for its reproductive function [28], but in accordance with the protection of the rights of children and adolescents [29, 30] and the protection of their 231 sexual freedom and sexual indemnity.

Response: It´s ok.

Comment 11. line 233 - take out 'the' - on one hand...

Response: Done.

Comment 12. line 277-278. remove dashes

Response: Done.

Comment 12. line 284 - result in rather than derive?

Response: In accordance with the comment suggested by the reviewer, the authors have changed "derive" to "result."

Reviewer 2 Report

Sexual abuse vs sexual freedom? A legal approach to the age of  sexual consent in adolescents in Spanish-speaking countries

The articles fall in between minor and major revision, the major revision draw attention to the limitation within the scientific background and introduction and the discussion around the lack of consensus on a definition on child and adolescent sexual abuse (CSA). The ‘Romeo and Juliet’ clause are not precisely explained, analysed, and included in the discussion in the introduction. Hence, I find the second objective complicated to understand since the Romeo and Juliet clause is not precisely explained. Further the article could be strengthened if the authors include philosophical scientific approaches for instance making use of Ian Hacking (1999) argument in the book Construction of What?  To elaborate on the precisely what is been constructed and what make up child and adolescent abuse. Emphasize both process and product in the way of constructing child abuse in relation to consent and what is understood as childhood and adolescence in the discussion on the legal age and consent. 

Author Response

Response: First of all, the authors welcome your comments and suggestions. The authors have included a brief additional explanation about the Romeo and Juliet clause. Therefore, we hope that the specific objective is now clearer. Regarding including other philosophical scientific approaches, such as the arguments of Ian Hacking (1999), it is really interesting. However, we have not focused in this research on this approach or on others, since the main objective is to review the minimum age of sexual consent and not so much to analyze what constitutes child and adolescent abuse. However, we will take it into account for future research.

Reviewer 3 Report

This is a very interesting and well developed article. However, I would like to make some minor suggestions.

Lines 105-107.

Therefore, sexual freedom is related to the assumptions of valid sexual consent, while sexual indemnity refers to the assumptions in which there is no legal capacity to consent and therefore, despite giving said consent, it would not be valid.

Some synonym?

Líneas 214-216

Finally, we discussed the relationship between the age of sexual consent and sexual freedom and sexual indemnity, as well as the protected legal rights in sexual crimes that involve minors and adolescents.

Unnecesary. I suggest to quit for an easier reading

Líneas 227-228

Age of sexual consent means the age below which, in accordance with national law, it is prohibited to engage in sexual activities with a child.

There is the same sentence in introduction and with a reference. Correct, please.

Line 258

abovementioned

above mentioned

References

I think they must be revised.

Author Response

Comment 1. This is a very interesting and well developed article. However, I would like to make some minor suggestions.

Response: The authors greatly appreciate the comments and suggestions made by the reviewer.

Comment 2. Lines 105-107.

Therefore, sexual freedom is related to the assumptions of valid sexual consent, while sexual indemnity refers to the assumptions in which there is no legal capacity to consent and therefore, despite giving said consent, it would not be valid.

Some synonym?

Response: Based on the reviewer's comment, the authors have changed the second "therefore" to "so that".

 Comment 3. Líneas 214-216

Finally, we discussed the relationship between the age of sexual consent and sexual freedom and sexual indemnity, as well as the protected legal rights in sexual crimes that involve minors and adolescents.

Unnecesary. I suggest to quit for an easier reading

Response: Done.

Comment 4. Líneas 227-228

Age of sexual consent means the age below which, in accordance with national law, it is prohibited to engage in sexual activities with a child.

There is the same sentence in introduction and with a reference. Correct, please.

Response: We, the authors, have modified the sentence and added the citation. Thank you for your comment.

Comment 5. Line 258

abovementioned

above mentioned

Response: Done.

Comment 6. References

I think they must be revised.

Response: Done

Reviewer 4 Report

 Sexual abuse vs sexual freedom? A legal approach to the age of sexual consent in adolescents in Spanish-speaking countries

Decision: Accept with minor changes

Introduction: It presents the issues very well and the definitions. However, there are some weaknesses.

On lines 97-100, the authors describe the "Romeo and Juliet" clause. It is essential to provide additional information about the importance of Romeo and Juliet for this manuscript. The suggestion is to elaborate more about the Romeo and Juliet clause.

The objectives of this study are well defined; however, it is not clear why the authors selected the Spanish-speaking countries to conduct this research? One of the recommendations is to show the literature gap in Spanish-speaking countries.

Discussion: Line 214-216: "Finally, we discussed the relationship between the age of sexual consent and sexual freedom and sexual indemnity, as well as the protected legal rights in sexual crimes that involve minors and adolescents." The recommendation for the authors is to remove this sentence.  Perhaps use it at the end of the discussion. At this stage, it isn't easy to follow the paragraph.

Limitations: The authors presented limitations well. Perhaps future studies could focus on the similarities and differences between Spanish-speaking countries and other countries.

Conclusions: The conclusions are intended to be concise. It is recommended to remove the paragraph lines 303-307.

Line 315, the authors use the word "states" that made the reader think they discussed the states of the US. However, they should be consistent and use the word "countries."

Author Response

The authors greatly appreciate the comments and suggestions made by the reviewer.

Introduction: It presents the issues very well and the definitions. However, there are some weaknesses.

Comment 1. On lines 97-100, the authors describe the "Romeo and Juliet" clause. It is essential to provide additional information about the importance of Romeo and Juliet for this manuscript. The suggestion is to elaborate more about the Romeo and Juliet clause.

Response: The authors have added the following information, in accordance with the suggestion made. "This clause is essential to avoid considering sexual activities among adolescents who have not yet reached the minimum age of sexual consent as criminal offenses. Otherwise, the sexualized and natural behavior of adolescent individuals would be penalized. However, the doctrine has also indicated that it is a necessary clause but with defects in its statement”.

Comment 2. The objectives of this study are well defined; however, it is not clear why the authors selected the Spanish-speaking countries to conduct this research? One of the recommendations is to show the literature gap in Spanish-speaking countries.

Response: The authors carried out an analysis of the legal system in Spanish-speaking countries due to the high prevalence of CSA in these countries, as we pointed out in the research. Another reason, precisely and according to your comment, is the lack of research in Spanish-speaking countries on the subject. Therefore, we have now included the following in the manuscript. “and the high prevalence of CSA in Spanish-speaking countries as well as the absence of investigations that have carried out a review of the legal systems of these countries in relation to the minimum age of sexual consent”

Comment 3. Discussion: Line 214-216: "Finally, we discussed the relationship between the age of sexual consent and sexual freedom and sexual indemnity, as well as the protected legal rights in sexual crimes that involve minors and adolescents." The recommendation for the authors is to remove this sentence.  Perhaps use it at the end of the discussion. At this stage, it isn't easy to follow the paragraph.

Response: Done.

Comment 4. Limitations: The authors presented limitations well. Perhaps future studies could focus on the similarities and differences between Spanish-speaking countries and other countries.

Response: Done. The authors agree with your comment and have included your suggestion. Thank you.

Comment 5. Conclusions: The conclusions are intended to be concise. It is recommended to remove the paragraph lines 303-307.

Response: The authors agree that the conclusions should be concise. However, the journal suggests that the authors remember the main objective and indicate the study sample. For that reason, we have not eliminated lines 303-307.

Comment 6. Line 315, the authors use the word "states" that made the reader think they discussed the states of the US. However, they should be consistent and use the word "countries."

Response: Done. The authors have changed "States" to "Countries".